# Investigation of the Failure Modes of Textile-Reinforced Concrete and Fiber/Textile-Reinforced Concrete under Uniaxial Tensile Tests

**DOI:** 10.3390/ma16051999

**Published:** 2023-02-28

**Authors:** Giorgio Mattarollo, Norbert Randl, Margherita Pauletta

**Affiliations:** 1Polytechnic Department of Engineering and Architecture, Università degli Studi di Udine, Via delle Scienze 206, 33100 Udine, Italy; 2Faculty of Civil Engineering and Architecture, Carinthia University of Applied Sciences (CUAS), Villacher Straße 1, A-9800 Spittal an der Drau, 9800 Carinthia, Austria

**Keywords:** textile-reinforced concrete (TRC), fiber/textile-reinforced concrete (F/TRC), textile fabric, short fibers, carbon textile, basalt textile, uniaxial tensile test, overlap length, pull-out

## Abstract

Recently, innovations in textile-reinforced concrete (TRC), such as the use of basalt textile fabrics, the use of high-performance concrete (HPC) matrices, and the admixture of short fibers in a cementitious matrix, have led to a new material called fiber/textile-reinforced concrete (F/TRC), which represents a promising solution for TRC. Although these materials are used in retrofit applications, experimental investigations about the performance of basalt and carbon TRC and F/TRC with HPC matrices number, to the best of the authors’ knowledge, only a few. Therefore, an experimental investigation was conducted on 24 specimens tested under the uniaxial tensile, in which the main variables studied were the use of HPC matrices, different materials of textile fabric (basalt and carbon), the presence or absence of short steel fibers, and the overlap length of the textile fabric. From the test results, it can be seen that the mode of failure of the specimens is mainly governed by the type of textile fabric. Carbon-retrofitted specimens showed higher post-elastic displacement compared with those retrofitted with basalt textile fabrics. Short steel fibers mainly affected the load level of first cracking and ultimate tensile strength.

## 1. Introduction

Textile-reinforced concrete (TRC) is a relatively new material, composed of a textile fabric and a cementitious matrix. In the literature, this material is also called fiber-reinforced concrete mortar (FRCM) and textile-reinforced mortar (TRM) [1]. Its main functions concern the retrofit of existing structures and the realization of new structural elements [2]. TRC became popular as an alternative solution to the well-known fiber-reinforced polymer (FRP) [3,4,5]. FRP is characterized by poor behavior above the glass transition temperature of epoxy resin, the high cost of epoxy resins, being hazardous for manual workers, non-applicability on wet surfaces and the incompatibility of epoxy resins with the substrate materials [3]. Such drawbacks are mainly related to the organic matrix [3,4,5]. Therefore, the use of a cement-based matrix instead of an organic one has led to the rise of TRC. Furthermore, TRC shares some advantages with FRP: the light weight to high strength ratio, the low impact on the original geometry and the corrosion resistance. For both components of the composite, the textile fabric and cementitious matrix, investigations to optimize the performance of TRC have been carried out in the last two decades. The effect of the matrix on the performance of TRC has been investigated by considering different types of cement-based matrices [6], high-performance concrete (HPC) [7,8] and ultra-high-performance concrete (UHPC) [9,10]. Furthermore, in order to improve the performance of cementitious matrices, lacking in terms of tensile strength, the admixture with short fibers has been studied. The use of short fibers in the TRC matrix has led to the creation of a new material called fiber/textile-reinforced concrete (F/TRC) [7,8]. The main advantages of admixing short fibers in the cementitious matrix are the increase in first crack stress and ultimate tensile strength and the improvement of crack control [8,10,11,12,13]. Concerning the textile fabric, different warps and materials have been studied, such as carbon, steel, glass, basalt, polypara-phenylene-benzobisoxazole (PBO) and aramid [14,15,16,17,18,19]. Among these materials, carbon is the most commonly used. The success of this material is mainly related to its high performance and light weight. However, among the materials available in the market, basalt is gaining increasing interest due to its mechanical performance and low environmental impact. In fact, the tensile stress and the elastic modulus of basalt fibers are generally higher than those of glass fibers and lower than those of carbon fibers [19]. Concerning the low environmental impact of basalt fibers, their production requires less energy than the production of carbon and glass fibers [20,21,22]. Furthermore, basalt fibers are characterized by high corrosion and temperature resistance. Basalt resists pH values up to 13 or 14, and shows good resistance to alkaline environments [19,23].

Despite the advantages of using a cementitious matrix, some drawbacks have to be considered. Due to the granularity of the mortar, the impregnation of the fibers of the rovings is very difficult to achieve [3]. This negatively affects the bond performance of the textile fabric with the matrix. The interaction between these two components, on TRC or F/TRC elements, is arduous to study if the properties of the textile fabric and the inorganic matrix are known separately. Furthermore, which of the different modes of failure that is most likely to happen cannot be easily estimated from the mechanical properties of the textile and cementitious matrix if studied separately. For this reason, tensile tests on composite elements are commonly executed to estimate their mechanical properties [9,24,25,26,27].

From the literature, it has been observed that, because of the large amount of materials available to realize textile fabrics and inorganic matrices, each combination of TRC system must be tested in order to characterize its tensile properties [27]. De Domenico et al. [6] conducted an experimental campaign of tensile tests on FRCM specimens. The materials used to realize the specimens were basalt, carbon and steel textile fabrics, short polymer fibers and three cementitious matrices characterized by compressive strengths of 22.03, 29.30 and 19.87 MPa. In this work, a comparison between the different configurations of the matrix and textile is carried out.

D’Antino and Papanicolaou [28] realized a mechanical characterization of TRM composites. They adopted a carbon textile fabric with and without coating, coated basalt textile fabric, coated glass textile fabric and galvanized steel textile fabric. Three different matrices were used, Matrix C, L and P, with compressive strengths of 16.40, 12.10 and 10.30 MPa, respectively.

De Felice et al. [29] executed an experimental campaign of uniaxial tensile tests and bond tests on brick and stone substrates. Carbon, basalt and steel textile fabrics were used with three different matrices: UNIRM3, a pozzolanic-cement mortar with a compressive strength of 37.0 MPa; UMINHO, a pozzolan lime-based mortar with a compressive strength of 13.0 MPa; and TECNALIA, a cementitious mortar with a compressive strength of 22.6 MPa.

Lignola et al. [30] conducted an extensive campaign of tensile and bond tests on FRCM specimens realized with three different basalt textile fabrics and four different inorganic matrices characterized by 25, 12, 15 and 15 MPa of compressive strength.

Larrinaga et al. [31], in order to propose an empirical non-linear approach to predict the stress–strain curve of TRM, executed an experimental test campaign on 31 specimens. The specimen utilized in this study was a basalt textile fabric covered by a bitumen coat. The matrix was a non-commercial cement-based mortar, characterized by a compressive strength of 19.8 MPa.

Hojdys and Krajewski [27] present the results of direct tensile tests on FRCM specimens characterized by three different textile fabrics (carbon, glass and PBO) and four types of inorganic matrix characterized by a compressive strength of 15.4, 14.9, 9.8 and 44.3 MPa, respectively. This study investigates the mechanical properties of FRCM systems, modes of failure and finding a bi- or tri-linear curve for the tensile stress–strain relationship.

Beßling et al. [32] developed a TRC/TRM system in which two inorganic matrices were investigated, a UHPC and a HPC matrix with a compressive strength of 83 ± 7 and 60 ± 5 MPa, respectively. Furthermore, carbon and basalt textile fabrics were used.

Zhou et al. [33] present an investigation about the effect of the reinforcement ratio, volume fraction of steel fibers and prestressing on Carbon TRM specimens under uniaxial tensile tests. The cementitious matrix adopted is a high-performed fine-grained mortar, characterized by a compressive strength of 76.7 MPa after 28 days. Short steel fibers characterized by a length of 12–15 mm and a diameter of 0.18–0.23 mm were admixed in the cementitious matrix. The textile fabric adopted is made by carbon fibers. A summary of the literature review presented above is listed in Table 1.

Among the research works presented above, in those of Beßling et al. [32] and Zhou et al. [33] similar solutions were used for the cementitious matrices of the tested specimens. In fact, Beßling et al. [32] used HPC and UHPC cementitious matrix and Zhou et al. [33] adopted a high-performed fine-grained mortar admixed with short steel fibers.

The inorganic matrices used by the authors differ in their composition and the presence or absence of short fibers, resulting in different mechanical performance, affecting aspects such as the compressive strength and the tensile strength. The textile fabrics differ from each other in the material of the constituent fibers, the shape of the rovings, the warping and the presence of a pre-coating. All of these aspects affect the behavior and performance of TRC, leading to each system being considered as unique.

## 2. Research Method, Content and Significance

By considering the state of the art in the TRC retrofit, the combination of high-performance concrete (HPC) admixed with short steel fibers with basalt and carbon textiles adopted in this work is not yet sufficiently investigated. For this reason, because the materials adopted here are commercial products, this work is an important contribution to the research and innovation of TRC and F/TRC. This is also a preliminary step of a larger experimental investigation on applications of these TRC and F/TRC systems on RC structural elements [34,35]. The results will be applied to refine in future research works the analytical formulations used to calculate the increase in performance of structural elements retrofitted with these TRC or F/TRC systems.

Finally, the main purpose of this study is to determine the tensile behavior and the mode of failure of these composite strengthening layers. The variables investigated are the material of the textile fabric (basalt and carbon), the influence of admixing short steel fibers in the cementitious matrix and the presence of an overlap of the textile fabric in the specimen. A total of 24 uniaxial tensile tests were conducted on specimens of Carbon-TRC (C-TRC), Basalt-TRC (B-TRC), Carbon-F/TRC (C-F/TRC) and Basalt-F/TRC (B-F/TRC) at the Science & Energy Labs of the Carinthia University of Applied Sciences in Villach (Austria).

## 3. Materials and Methods

The cementitious matrix of the specimens was obtained as a fine-grained premix approved in Germany to be used as a cementitious matrix in TRC retrofit [36]. Short steel fibers with a length *l* of 5 mm and a diameter *d* of 0.15 mm were admixed in the cementitious matrix in order to be used in F/TRC. An amount of 2.5 vol.-% of short steel fibers was admixed in the cementitious matrix. Through compressive tests on 100 × 100 × 100 mm^3^ cubes and a splitting tensile test on 100 × 200 mm^2^ cylinders, the compressive and splitting tensile strength *f_c_* and *f_t_* of the cementitious matrix are approximately 93.6 and 3.6 MPa, respectively. In the fiber-reinforced matrix, the compressive and tensile strengths are, respectively, 105.2 and 10.9 MPa. The following geometrical and mechanical properties of the textile fabrics are declared by the producers [37,38,39]. The grid opening *g* of the basalt textile fabric is 20 × 20 mm^2^ and the cross sectional area *A* per meter is 65 mm^2^/m. The elastic modulus E, the tensile strength *σ_t,tex_* and the strain at the maximum load *ε_u_* of basalt textile fabric are 92.7 GPa, 1495 MPa and 1.61% in both directions, respectively. The carbon textile fabric is characterized by a grid opening *g* of 22 × 22 mm^2^, and the cross-sectional area *A* per meter is 71 mm^2^/m. The tensile strength *σ_t,tex_* and strain at the maximum load are 2531 MPa and 1.71% lengthways and 2841 MPa and 1.47% crosswise, respectively. The properties of the cementitious matrices and the textile fabrics are presented in Table 2 and in Table 3, respectively.

The experimental test campaign included 24 specimens divided into two series: A and B. Series A concerns specimens composed of one layer of basalt or carbon TRC and F/TRC to be tested under the uniaxial tensile test. The specimens are 120 mm wide, 15 mm thick and 600 mm long. The textile fabric is placed halfway through the thickness of the specimen (Figure 1). The design of the specimens is in accordance with the recommendation of RILEM TC 232-TDT [24]. The aim of testing specimens of series B is to investigate the resistance of the textile fabric overlap in one layer of TRC or F/TRC. In this series, the specimen’s dimensions differ from those of series A in terms of length (810 mm for the specimen with basalt textile fabric and 710 mm with carbon textile fabric) and thickness (on carbon- and basalt-retrofitted specimens, the thickness is close to 18 and 16 mm, respectively). The overlap length is 150 mm and 250 mm on carbon and basalt textile fabric, respectively. Two pieces of textile fabrics, overlapped in the center of the specimens (see Figure 1), are placed in each specimen. The overlap is realized with two pieces of textile 430 mm and 530 mm long, overlapped in order to obtain an overlap length of 150 mm and 250 mm on carbon and basalt textile fabric, respectively. These values refer to the overlap length suggested by the producers, which is increased by 50 mm in order to avoid premature failure in the overlap zone. The reinforcement ratio ρ = Atextile/Amatrix is 0.5% for C-TRC and C-F/TRC and 0.4% for B-TRC and B-F/TRC specimens (Figure 1).

The first step for the preparation of the specimens is the realization of the wooden formworks, which are 6 mm-thick rectangular frames suitable for the realization of a 6 mm-thick homogeneous layer of cementitious matrix, as shown in Figure 2a. The next step is to apply the textile fabric to the fresh matrix and fix it on the two opposite sides to the formwork in order to keep it tight and maintain the position during the application of the last layer of cementitious matrix. Finally, the last 6 mm-thick layer of cementitious matrix is applied as a cover (Figure 2b–e). The thickness of the textile and the clips used to fix it is approximately 3 mm. The fabrication process for specimens of series B differs only during the phase of application of the textile fabric. Two pieces of textile fabric are overlapped: once the first 6 mm thick layer of cementitious matrix is applied, the first piece of textile fabric is placed and fixed along the lateral sticks of wood with clips; subsequently, wood sticks are placed laterally to the specimen in order to act as a guide for the application of a 2 mm-thick layer of cementitious matrix; then, the last piece of textile fabric is applied and fixed. Finally, the last 6 mm of cementitious matrix is applied as a cover. 

The test setup, shown in Figure 3a, refers to the recommendations of RILEM TC 232-TDT [24], Digital Image Correlation system was used to measure strains instead of the traditional strain transducers. To perform the tests, the two ends of the specimens are placed inside two steel clamps of the test machine (Figure 3b). The load is applied on specimens through friction by tightening the six bolts present for each clamp. A thin layer of rubber is placed between the steel clamps and the specimen to improve the adherence between steel and the cementitious matrix.

The steel clamps are 180 mm long, resulting in a free length of 240 mm on uniaxial specimens of series A. On specimens of series B, the free length is 350 and 450 mm for specimens with carbon and basalt textile fabric, respectively. The age of the cement-based matrix during the test phase is between 211 and 224 days on specimens with matrix not admixed with short steel fibers, and between 240 and 252 days for specimens with short steel fibers admixed in the matrix.

Specimens are aligned with the jaws in order to apply pure tension to the specimen. Tests are performed under displacement control with a speed of 0.5 mm/min. The tests are also measured with the Digital Image Correlation system (DIC), which is appropriate to survey the cracking behavior. Furthermore, the axial displacement of the machine was measured during the tests through the DIC. Through the analysis with the DIC software it is possible to select two points in the chosen area and to measure the displacement of these points during the test. These points together are referred to in this article as the “virtual extensometer”. The axial displacement imposed by the test machine during the test is measured through a virtual extensometer by placing two points in the middle of the two steel clamps, one on the top and one on the bottom of the test machine. The measurement of crack opening and deformation are obtained by placing three virtual extensometers across the crack that propagate in the specimen. Therefore, the two virtual points are placed on the right and on the left of the measured crack. Crack deformation values are obtained by dividing the crack displacement by the initial displacement.

## 4. Results

Table 4 summarizes the main results of the tests: maximum load, equivalent yielding displacement δ_y_, ultimate displacement δ_u_ and index of inelastic displacement µ. The index of inelastic displacement is the ratio between the ultimate displacement and the equivalent yielding displacement, which expresses the capacity of the specimens to undergo increasing displacement after the first elastic branch to a certain extent before failing. The ultimate displacement is considered here as the displacement corresponding to a decrease of 20% in the ultimate load of the descendant curve [40]. The equivalent yielding displacement corresponds to the displacement occurring at the first transversal crack. In Table 4, values of the coefficient of variation (CV) are listed. While values of maximum load are characterized by low CV, the index of inelastic displacement is generally characterized by a higher CV. Firstly, μ is the ratio between δ_u_ and δ_y_; hence, phenomena that influence these two parameters are taken into consideration. For TRC specimens in particular, the values of δ_y_ are low; therefore, even slight variations in δ_y_ have a significant impact on μ values and may increase the CV of μ. The study of crack propagation in the specimens is limited by the fact that, during the tests, the test machine steel clamps cover the ends of the specimens. These areas are not measured by the DIC systems; thus, the propagation of cracks cannot be observed there. The propagation of cracks within the steel clamps would affect the total displacement capacity of the specimens, and consequently the ultimate displacement δ_u_ and μ. Finally, minimal geometric imperfections of the textile fabric and imprecision during the specimens’ production process may increase the variability of the test results.

The ID used to identify the specimens consists of a code in which the first letter refers to the material of the textile fabric (B for basalt and C for carbon), the second letter refers to the absence or presence of short steel fibers in the cementitious matrix (P is without fibers and Pf is with fibers), the number identifies one of the three identical specimens and the last letter is the name of the series (series A or series B). For example, the ID BPf-3-A refers to a specimen retrofitted with basalt textile embedded in a cement-based matrix admixed with short steel fibers, the third specimen of series A.

### 4.1. Series A

#### 4.1.1. B-TRC vs. B-F/TRC

Force–displacement and stress–strain curves of B-TRC (in red, in Figure 4a,b) can be simplified as a bilinear curve characterized by a stiff linear first branch and a second branch characterized by a lower stiffness, which lasts until the maximum load. The specimens fail abruptly after the maximum load is reached. The average maximum load for the B-TRC specimens is 6178.3 N. The change in stiffness between the two linear branches is approximately situated at a load of 3000 N in the force–displacement curve. Alongside this changes of stiffness, the first crack appears, followed by a second crack that appears during the development of the second branch. The deformations of the specimens are mainly concentrated in the widening of one crack.

From Figure 5 and Figure 6, it is possible to observe that, at failure, the main crack propagates from one side to the other of the specimen. In these pictures, colors are scaled to represent the major principal strain values. Figure 7 shows six groups of graphs. In each group, the first two graphs refer to the measurement of the main crack opening, and the third is the force–displacement curve of the specimens during the test. Crack opening is expressed through three curves obtained by three virtual extensometers positioned to measure the crack opening; two are placed at the ends and one in the center of the specimens (see Figure 5). In the legend of Figure 7, E1, E2 and E3 represent the three virtual extensometers. In these graphs, the vertical axis corresponds to the displacement between the two points that define the virtual extensometer, while the horizontal axis corresponds to the axial displacement of the two steel clamps during the test. Finally, one last virtual extensometer is applied on the steel clamps, to measure the displacement imposed by the test machine. The force–displacement curve is obtained by displaying the force measured by the test machine versus the axial displacement measured by the virtual extensometer. In each group, graphs are vertically aligned to observe how the crack opening behaves compared with the force–displacement curves. In line with the maximum load, cracks tends to open abruptly, followed by a sudden decrease in load (see Figure 7a,c,e). From the curves of the virtual extensometers, it can be observed that the main crack tends to widen on one side and to close on the opposite side of the specimen. Therefore, in all specimens, the crack opening is not uniform along the transverse section. This produces specimen rotation, as can be seen in Figure 5.

By observing the failure conditions for the basalt textile B-TRC specimens, it appears that the failure of the specimens occurred due to the rupture of the textile after the principal crack (Figure 8a).

The test results of B-F/TRC specimens are represented in Figure 4 in terms of force–displacement curves. The behavior of these curves can be simplified into three parts: a first, almost linear behavior, a plateau, and a descendant branch which leads to the failure of the specimen. The first branch is characterized by a linear increase in the load until approximately 9000–10,000 N. Once this phase is reached, a decrease in stiffness follows, leading to a plateau in the force–displacement curves. During the degradation of stiffness and until the end of the plateau, a progressive propagation of transversal cracks develops along the specimens (Figure 9a–c). At the end of the plateau, the progressive drop in load results in the concentration of the imposed displacement in the widening of one principal crack (Figure 9c). During the decrease in load, at approximately 6000 N, the main crack is approximately 2 mm wide and expands suddenly at failure. During the failure of the specimens, it has been observed that the main crack tends to widen more on one side than the other, which results in a “rotation” of the specimen during the failure. This observation is confirmed by the crack opening curves in Figure 7b,d,f, in which the curves of the virtual extensometers show an increase in displacement (or deformation) on one side of the specimen and a decrease in the other side.

The mode of failure of B-F/TRC specimens is the same as that of B-TRC specimens: rupture of the basalt textile fabric. The possibility of the pull-out of the textile fabric is excluded by observing the failed specimens. Figure 8b–d show the main crack of specimen BPf-1-A and the details of the textile at the two ends, showing no apparent slippage of the rovings.

#### 4.1.2. C-TRC vs. C-F/TRC

The results of the tests are represented by force–displacements and stress–strain curves in Figure 10a,b.

The first stage is characterized by high stiffness, governed by the interaction between the matrix and the textile fabric. This first linear behavior lasts until the load reaches approximately 4000–5000 N, where the first transversal crack appears on the cementitious matrix. The second stage is characterized by lower stiffness and a behavior which could be approximated as linear in the first instance. During the increase in load, the displacement imposed in the test is mainly concentrated in the widening of the crack. Once the maximum load is reached, a drop in load of approximately 2000–3000 N characterizes the third stage. This leads to the fourth stage, in which the load slightly increases asymptotically to approximately 8000 N, resulting in a pull-out failure. This is confirmed by checking the details of the cracks in Figure 11 and the graphs in Figure 12a–c, which show that cracks, at the last stage of the force–displacement behavior, tend to open continuously at almost constant load. This is expressed by the crack opening curves in which the virtual extensometer, after the drop in load, proceeds almost linearly.

The mode of failure of C-F/TRC specimens is governed by the pull-out of the textile fabric from the cementitious matrix; however, the shape of the force–displacement curve slightly differs from that of the C-TRC specimens in the first and second stage. As shown in Figure 10, the maximum load is generally reached in the first stage, while the second branch is descendant. The average maximum load is 12,159 N. In close proximity to the maximum load, a first transversal crack appears, whereas the stiffness is reducing. This event leads to the second stage, which is characterized by a decrease in the load and a progressive propagation of transversal cracks along the specimen. The propagation of transversal cracks along the specimen is shown in Figure 13. In this phase, the presence of the short steel fibers affects the performance of the layer by reaching higher loads in the various stages and by distributing stresses along the specimens through the effect of the short fibers. In the third stage, the drop in load also constitutes a quick widening of the crack, leading to the last part of the graph: an almost linear behavior in which the load slightly increases due to the friction between the rovings of the textile fabric and the cementitious matrix. This is also observed in Figure 12d–f: once the maximum load is reached, the displacement measured by the virtual extensometers increases almost linearly until the end of the test.

### 4.2. Series B

#### 4.2.1. B-TRC vs. B-F/TRC

The test results of the B-TRC specimens of series B are represented by force–displacement and stress–strain curves in Figure 14a,b.

Generally, the behavior of these tests is similar to that of series A. In fact, the force–displacement curves are characterized by a similar bilinear behavior. The linear branch lasts until approximately 4000–5000 N. At the end of the first linear branch, the first crack appears and there is a sudden decrease in stiffness. In the second branch, the load increases—with a reduced stiffness—until the maximum is reached. The average maximum load is 7290 N, which is 18% higher compared with that of series A. The failure mode of specimens of series B is the same of those of series A: rupture of the textile fabric. Once the maximum load is reached, the force instantly decreases and the principal crack simultaneously widens. Regarding the mode of failure of B-TRC specimens in series A, the propagation of the principal crack starts from one side and propagates transversally through the specimen. In Figure 15, a comparison between the force–displacement curves and the crack opening–axial displacement of the main crack is presented for each B-TRC and B-F/TRC specimen. The behavior of the virtual extensometer’s curves is similar to that of series A for both B-TRC and B-F/TRC specimens. Once the maximum load is reached, the displacement measured on one side of the specimen starts to increase, while on the other side it tends to decrease, resulting in a slight rotation of the specimens around the main crack.

The behavior of the force–displacement curves of B-F/TRC specimens of series B is similar to that of series A in terms of average maximum load and shape of the curves. Four stages characterize the force–displacement curves. Firstly, a linear branch lasts until 10,000–11,000 N. Due to the formation of the first transversal cracks, the stiffness decreases, leading to the second branch in which the load reaches its maximum values and begins to decrease. The third stage, related to the decrease in the load coincides with the widening of the principal crack. During the widening of the principal crack, there is the pull-out of the short steel fibers. During this phase, the load tends to decrease and to intercept the curves of B-TRC specimens. At approximately 7000–8000 N, the curves tend to intercept those of B-TRC specimens, leading to a final drop in load corresponding to the rupture of the basalt textile fabric. The mode of failure is still characterized by the non-homogeneous propagation of the crack.

#### 4.2.2. C-TRC vs. C-F/TRC

In terms of force–displacement and stress–strain curves, the behavior of the C-TRC specimens of series B, as shown in Figure 16a,b, is characterized by a first linear step, which lasts until approximately 4000–5000 N. Corresponding to the appearance of the first crack, the stiffness starts to decrease until the axial displacement is 2.5 mm and the load is approximately 8000 N. A drop in load precedes the last stage of the curves, corresponding to the tendency of the curves to have large displacements under a reduced increase in loads. The mode of failure of these specimens is similar to that of specimens of series A: the pull-out of textile fabric from the cementitious matrix. It can be seen from the results of these tests that the force–displacement behavior is slightly different compared with that of specimens of series A.

In Figure 17, a comparison between the force–displacement curves and the crack opening–axial displacement of the main crack is presented for each C-TRC specimen of series B. In this series, specimens are characterized by two main cracks. Therefore, several crack opening and crack deformation graphs are presented. Similarly to what happens in series A, the displacement measured by the virtual extensometer tends to increase with similar behavior, which is consistent with the pull-out of the textile fabric from the cementitious matrix. The two main cracks influence each other’s behavior; when the rate of widening of the first crack decreases, that of the second one increases and vice versa.

The force–displacement curves of C-F/TRC specimens of series B are characterized by behavior similar to that of series A. A first branch, which lasts until approximately 14,000 N, corresponds to the formation of the first cracks and the consequent decrease in stiffness. This leads to the maximum of the curve, in which the maximum load is reached. The maximum load is higher in these series of tests since specimens are slightly thicker compared with those of series A: approximately 18 mm instead of 15 mm (20% higher). Consequently, in the next stage, the load decreases until approximately 12,000–13,000 N. What follows is the loss of the bond between the textile fabric and the cementitious matrix, resulting in a drop in load of approximately 4000 N. The last stage of the curves is related to the pull-out of the rovings from the cementitious matrix. In Figure 18, during the second stage, at maximum load it can be observed that the principal cracks tends to widen. There, the crack proceeds by widening quite homogenously until the drop in load of the fourth stage. This is observable in the crack opening curves, where the displacement measured by the virtual extensometer increases similarly in all three curves.

## 5. Discussion

### 5.1. B-TRC vs. B-F/TRC Series A

The behavior of the B-TRC specimens of series A is characterized by a first linear branch which is influenced by both the matrix and the textile fabric. After the first crack, the load is spread over the basalt textile fabric. During the failure, the principal crack tends to widen non-uniformly: one side of the crack tends to widen more, resulting in a “rotation” of the specimen during the test (Figure 6c). This suggests that the single rovings are not equally loaded at failure. This “rotation” is confirmed by the crack opening–axial displacement behavior (Figure 7a–e). The virtual extensometer applied on the crack shows that on one side the crack widens abruptly, while on the other side it tends to close. By assuming that the activation of the rovings is proportional to the displacement at the failure of the crack, it is possible to estimate the load at failure. For simplicity, a linear stress–strain distribution is considered along the principal crack (Figure 19).

This behavior is based on the assumption that the displacement along the crack develops linearly, where one side is fully activated and the other is not activated. The rovings in the middle are activated proportionally according to the linear behavior. This phenomenon, which leads to a non-uniform distribution of stresses through the textile fabric, may be due to the natural non-uniformity of the cementitious matrix, minimal geometric imperfections of the textile fabric and imprecision during the production process. Considering that the average tensile strength of the roving is approximately 1495 MPa [38] and the cross-section area of one roving is 1.3 mm^2^, the theoretical maximum load of six rovings equally activated is approximately 11,661 N. Nevertheless, by assuming an activation of the textile fabric, with a linear activation of the rovings, the maximum load is approximately 5830 N, 5.6% lower compared with the average maximum load of the experimental results, 6178.3 N.

On B-F/TRC specimens of series A, the behavior of the distribution of stresses is affected by the presence of the short steel fibers in the cementitious matrix. Compared with B-TRC specimens, the higher load of the first branch is to be attributed to the short steel fibers admixed in the cementitious matrix, which improves the performance of the cementitious matrix in terms of tensile strength. With a reduction in stiffness the first crack appears, and a pattern of transversal cracks spreads along the specimens. This is a typical behavior related to the short fibers admixed in the matrix. During the propagation of cracks along the specimens, the principal crack, which widens abruptly during the failure of the specimen, seems to open abruptly with a crack width of 1.5–2.0 mm. During the widening of the principal crack, characterized by the pull-out of fibers, the load decreases, intercepting the curves of specimens B-TRC. In this phase, the load is increasingly transmitted to the basalt textile fabric. The load decreases progressively until approximately 6000 N; then, when the force–displacement curves of B-F/TRC specimens intercept the curves of B-TRC specimens, the textile fails due to the rupture of the basalt fibers. At this point, the mode of failure is similar to that of B-TRC specimens and the maximum load is mainly attributed to the presence of the short steel fibers in the cementitious matrix.

Concerning the average index of inelastic displacement µ_av_ in Table 4, it appears that the B-F/TRC specimens show a reduction of approximately 93.8% in µ_av_ compared with specimens of B-TRC. This might suggest that B-F/TRC is characterized by a poor post-elastic behavior compared with B-TRC specimens. However, it seems that the force–displacement curves of B-F/TRC specimens tend to intercept those of B-TRC during the descendant branch. This leads us to believe that only considering the value of µ may not fully describe the post elastic capacity of the material. In fact, the failure of both B-TRC and B-F/TRC is related to the rupture of the textile fabric, with similar values of load and axial displacement. This leads to the observation that the actual axial displacement capacity before failure is comparable on both materials.

### 5.2. C-TRC vs. C-F/TRC Series A

The curves of tests on C-TRC specimens can be simplified by the multilinear curve in Figure 20a, which, apart from the first stage, is comparable to that obtained by Ortlepp et al. [41] from pull-out tests. The first stage of the multilinear curve lasts until the first crack appears and a consequent reduction in stiffness is observed (stage “1”). In the second stage, while reaching the maximum load, the adhesion between the textile fabric and the cementitious matrix is progressively activated and the first transversal crack initially formed is usually followed by the propagation of a second crack (stage “2”). In the third branch, the bond between the carbon textile fabric and the cementitious matrix is lost, which results in a reduction in the load in the force–displacement curve (stage “3”). Finally, in the fourth stage, the load slightly increases asymptotically to approximately 8000 N, resulting from the friction between the textile fabric and the cementitious matrix, leading to the pull-out of the textile (stage “4”). In the fourth stage, as shown in Figure 10, the curves are characterized by small drops in force, probably caused by the manner of widening of the crack: the widening is not uniform along the crack, but it is always more pronounced on one side, and after a small drop, it is more pronounced on the other side, as shown Figure 20b.

The force–displacement curves of the C-F/TRC specimens can be simplified and approximated with a multilinear behavior (Figure 21). The multilinear behavior seen in Figure 20 and Figure 21 differs mainly in the first two stages. For C-F/TRC specimens, the maximum load is reached in the last part of the first almost linear stage. While approaching the last part of this branch, the first crack appears, leading to a reduction in stiffness (stage “1”). In the second stage of the curve, the load, differently from C-TRC curves, decreases until approximately 10,000 N. The load reached at the end of the second stage is close to that observed for C-TRC specimens. During this stage, it is assumed that, as for the C-TRC specimens, the textile fabric is progressively activated (stage “2”). Finally, the third and fourth stages are comparable to those of C-TRC specimens. The drop in load is attributed to the loss of bond between the rovings of the textile fabric and the cementitious matrix (stage “3”). The long final linear branch is caused by the friction between the rovings and the cementitious matrix during the pull-out of the textile fabric from the cementitious matrix.

Figure 12 shows the tendency of the principal cracks to open uniformly along the transversal direction. This information could imply that the contribution of each roving during the widening of the crack is similar.

Due to the definition of the index of inelastic displacement µ given here, the large increase in axial displacement at almost constant load characterized by the pull-out failure is not considered. However, µ expresses how C-TRC specimens realize larger axial displacement between the first crack and a drop of 20% in the maximum load.

Both C-TRC and C-F/TRC specimens failed with the pull-out of the textile fabrics from the matrix. This is a premature failure of the specimens, since the maximum tensile capacity of the carbon rovings has not been exploited. In fact, the maximum load that five rovings could carry is approximately 25,860 N, since the tensile transverse strength is 2841 MPa and the cross-section of each roving is 1.82 mm^2^ [37]. The average maximum load reached by the C-TRC and C-F/TRC specimens is 10,296.7 and 12,159 N, respectively, 60% and 53% lower compared with the potential maximum load of the total of the five rovings of the specimens. This type of failure is caused by the insufficient bond between the textile fabric and the matrix, which prevents the rovings from reaching their maximum capacities. As previously mentioned, this eventuality could arise when the textile fabric and the cement-based matrix cannot impregnate the inner fibers of the roving. The degree of penetration of the matrix and the shape of the cross-section of the rovings are salient to the bond connection. Flat rovings, such as that of the basalt textile, have a better distribution of stress [26], while on the contrary, the carbon rovings adopted here are characterized by a circular-cross section, which possibly leads to a worse penetration of the matrix. However, it must be pointed out that the higher declared tensile strength of the carbon textile fabric, compared with that of the basalt textile fabric, may also have contributed to making pull-out failure the main mode of failure.

The considerations above could explain the different modes of failure between the specimens with carbon and basalt textile fabric.

### 5.3. B-TRC vs. B-F/TRC Series B

B-TRC and B-F/TRC specimens of both series A and B are characterized by similar force–displacement behavior. This may be due to the fact that the overlap length of specimens of series B is long enough to avoid premature failure and leads to the same mode of failure: rupture of the basalt textile. However, the average maximum load of specimens of series B is higher compared with that of series A.

On B-TRC specimens of series B, the difference between the experimental average maximum load and the hypothesized linear distribution is approximately 20%. This high difference may be attributed to the tendency of cracks to open more uniformly compared with those of series A. This would lead to a slightly better distribution of stresses between the rovings at failure.

### 5.4. C-TRC vs. C-F/TRC Series B

The different behavior between C-TRC specimens of series A and B could be explained by the formation of cracks closer to the extremity of the specimens. Specimens of series B are characterized by cracks that are situated at approximately 150 mm from the edge, while for series A this length is bigger, being 177 mm on average. This leads to a bond length that is approximately 20% shorter. This could be explained by the fact that due to the presence of the overlap, cracks are formed closer to the edges of the specimens. The average maximum load of specimens of series B is 8905.2 N and of series A 10,375.1 N, which is approximately 15% higher. This could be explained by the lower force transmitted between the cementitious matrix and the textile fabrics because of the lower bond length. Consequently, the drop in force in the third stage is lower for the same reason, related to the reduced bond length. Specimen CP-2-B is characterized by higher loads, probably due to the fact that the position of the crack leaves a bond length of 200 mm from the border of the specimens, similarly to specimens of series A.

As for specimens C-F/TRC of series B, observations similar to those for series A can be made: the mode of failure of specimens retrofitted with carbon textile fabric is still the pull-out of the textile fabric. The presence of overlap does not affect the mode of failure of the composite.

In this experimental test campaign, two different modes of failure have been observed: rupture of the textile fabrics and pull-out of the textile fabric from the cementitious matrix. The latter failure mode, observed here on C-TRC and C-F/TRC specimens and related to the insufficient bond between textile fabrics and cementitious matrix, has also been observed by Hojdys et al. [27]. By comparing his results and those of other articles [30,42,43,44,45,46], he observed that this mode of failure happens especially when a clamping system with bolted steel plates is used. He suggests that, to provide adequate length for the textile fabric, the test setup could be modified. The suggestions proposed in his article include the use of pneumatic or hydraulic gripping to increase the pressure of the clamps on the specimen, designing longer specimens and longer steel plates for the clamping system and lengthening the specimen outside the bolted steel plates. While the force–displacement behavior of B-TRC specimens is similar to the classic trilinear behavior of TRC specimens under tensile loads, the force–displacement (and stress–strain) behavior of C-TRC specimens, affected by the loss of bond between textile fabric and cementitious matrix, is similar to that observed by Ortlepp and Lorenz [41]. In her work, Ortlepp executed pull-out tests on TRM specimens characterized by a high-strength concrete matrix and a textile fabric composed by carbon filament yarns in warp directions and alkali-resistant glass fibers in weft direction.

The influence of admixing of short fibers in the cementitious matrix of TRM has been investigated by Zhou et al. [33]. In his study, 2% by volume of short steel fibers were admixed in the cementitious matrix. The tensile strength of the TRM plates increases in approximately 100% compared to the reference specimens. A similar result has been obtained in this work, in fact an increase of approximately 83% and 92% of the average maximum load has been observed on B-F/TRC of series A and B, respectively. On C-F/TRC specimens, the increase in tensile strength was lower, 18% and 76% on series A and B, respectively.

Finally, the considerations about the results reported in here are limited by the materials, the specimen’s layouts and the test setup adopted. Therefore, because of the uniqueness of the combination of materials and the design of the specimens adopted here, results such as maximum tensile strength, and failure mode are specific to this work.

## 6. Conclusions

In this work, an extensive experimental test campaign was conducted in order to investigate the properties of textile-reinforced concrete (TRC) under tension. A total of 24 uniaxial tensile tests were conducted on specimens comprising one layer of carbon or basalt textile fabric immersed in a layer of inorganic matrix. Additionally, the influence of admixing short steel fibers in the cementitious matrix was investigated, which led to a new material called fiber/textile-reinforced concrete (F/TRC). Finally, the investigation involved the study of the behavior of TRC and F/TRC on the overlap of the textile. The results of the experimental test campaign are expressed in terms of force–displacement curves, stress–strain curves and crack opening–displacement curves. Due to the great variability of different textile fabrics, inorganic matrices and short fibers, each TRC and F/TRC system is unique and warrants further investigation. In this study, commercial high-performance concrete, short steel fibers and basalt and carbon textile fabrics are combined to realize unique TRC and F/TRC systems. Therefore, given the originality of the combinations of materials adopted here, an investigation on the mechanical properties of TRC and F/TRC is necessary. The results of this research will be used to better comprehend the behavior of existing reinforced concrete (RC) structures retrofitted with these systems. Furthermore, this is a fundamental step to refine the analytical formulations used to estimate the contribution of these systems to the increase in performance of the retrofitted elements.

From the analysis of the experimental test results, the following conclusions can be drawn:Carbon-retrofitted specimens perform better, in terms of maximum tensile load, compared with those retrofitted with basalt textile fabric. In this experimental test campaign, the average maximum loads are 66.7% and 22.2% higher for TRC specimens and 7.5% and 11.9% higher for F/TRC specimens of series A and B, respectively.Short steel fibers improve the performance of both basalt- and carbon-retrofitted specimens. The propagation of transversal cracks, which causes the reduction in stiffness of the first branch of the force–displacement curve, occurs at higher levels of load. The range of loads in which the stiffness of the force displacement curves decreases due to the propagation of transversal cracks increases from approximately 3000–4000 N to 9000–10,000 N in series A and from approximately 4000 N to 12,000–13,000 N in series B.Ductile failure characterizes carbon-retrofitted specimens. In this experimental test campaign, C-TRC and C-F/TRC specimens fail due to the pull-out of the textile fabric from the cementitious matrix. During the pull-out of C-TRC specimens, the average load is 85% and 84% of the average maximum load for series A and B, respectively. For C-F/TRC specimens, the average load during the pull-out of the textile fabric is approximately 72% and 60% of the average maximum load of series A and B, respectively.The assumed overlap length is long enough to avoid premature failures during the tests. The mode of failure of specimens with overlap of the textile fabric is similar to that of specimens without overlap of the textile fabric.A simple method to estimate the load capacity of basalt composites is proposed. The difference between the theoretical and experimental results is lower than 5.6% and 21% for series A and B, respectively.

## Figures and Tables

**Figure 1 materials-16-01999-f001:**
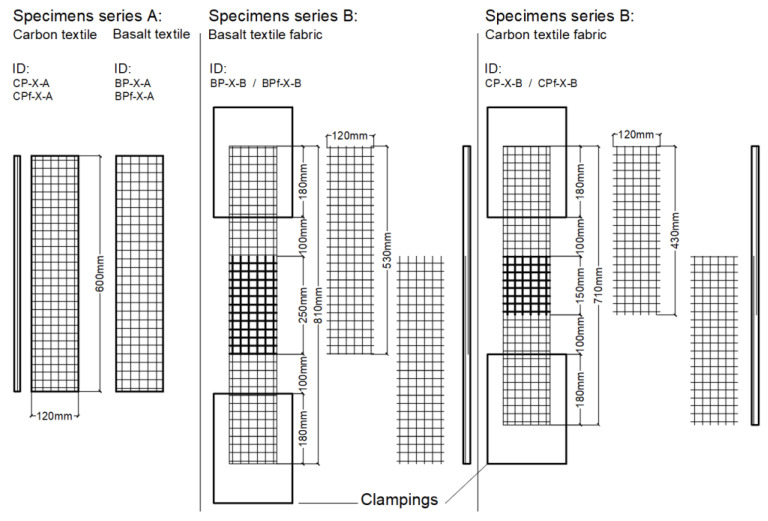
Details of the design for specimens of series A and series B.

**Figure 2 materials-16-01999-f002:**
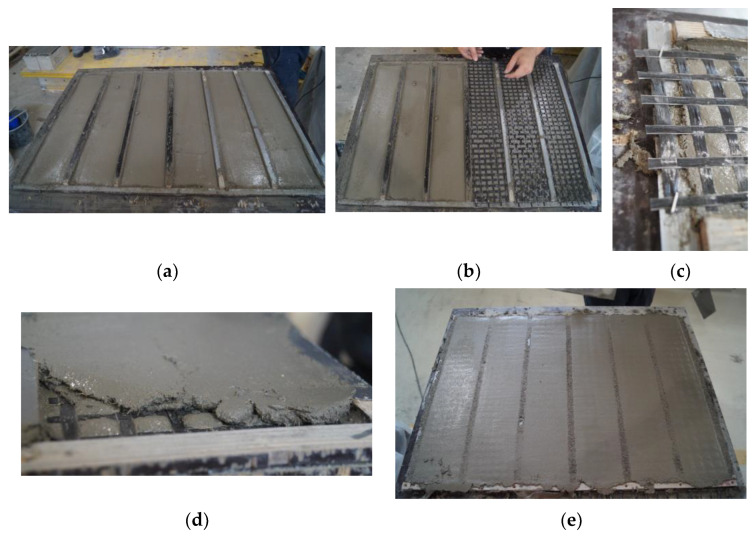
Details of the design for specimens of series A and series B. In (**a**), the first layer of cementitious matrix is applied; in (**b**), the basalt textile fabric is placed over the cementitious matrix; in (**c**), is shown a detail of how textile fabric is fixed on the wooden formwork; in (**d**), the last layer of cementitious matrix is applied; and in (**e**) is shown the specimens when the production process is concluded.

**Figure 3 materials-16-01999-f003:**
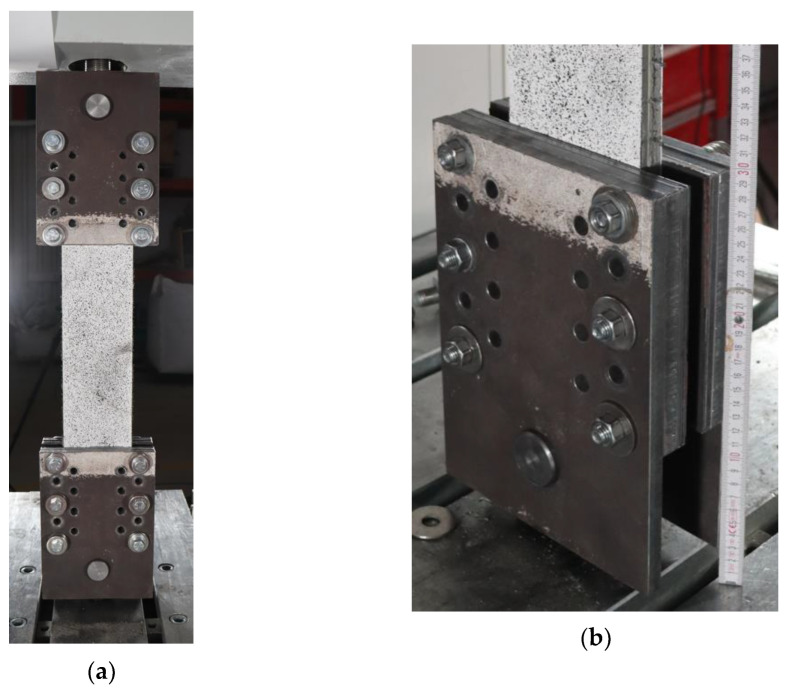
Specimen clamped in the testing machine (**a**), and a detail of the steel clamp (**b**).

**Figure 4 materials-16-01999-f004:**
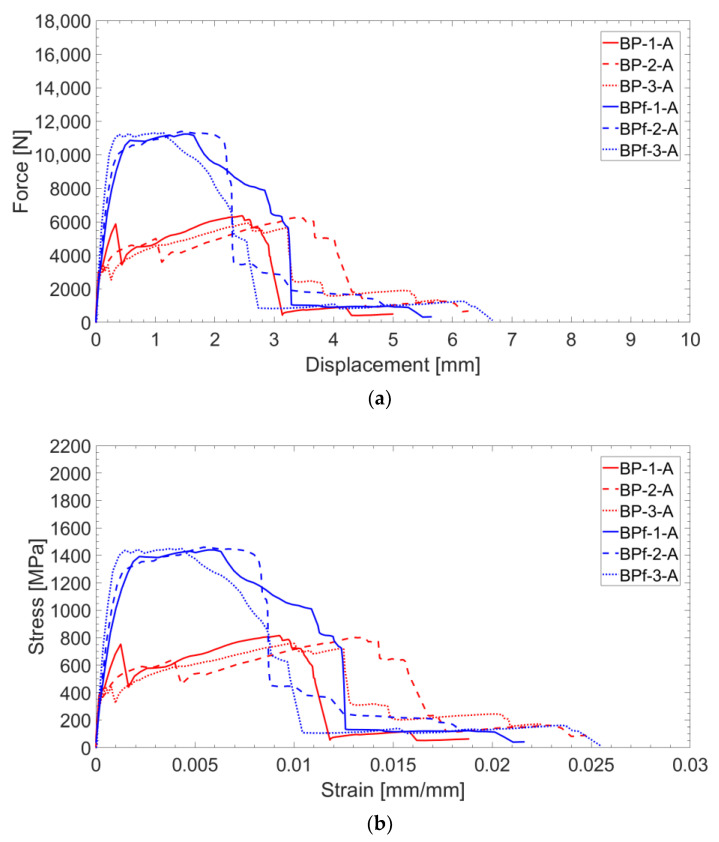
Force–displacement (**a**) and stress–strain (**b**) curves of B-TRC and B-F/TRC specimens.

**Figure 5 materials-16-01999-f005:**
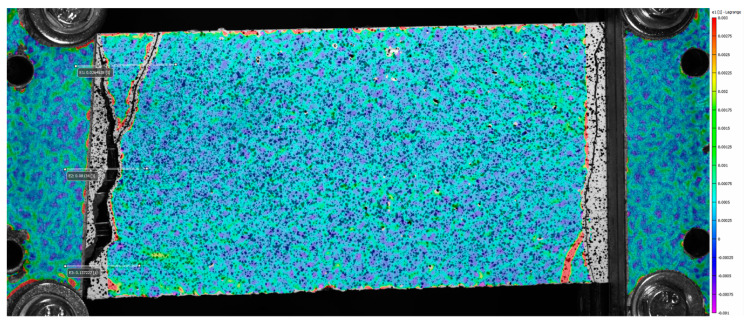
The virtual extensometer on the BP-A specimen. The color scale represents the grade of the major principal strains, from 0.003 units in red to −0.001 units in purple.

**Figure 6 materials-16-01999-f006:**
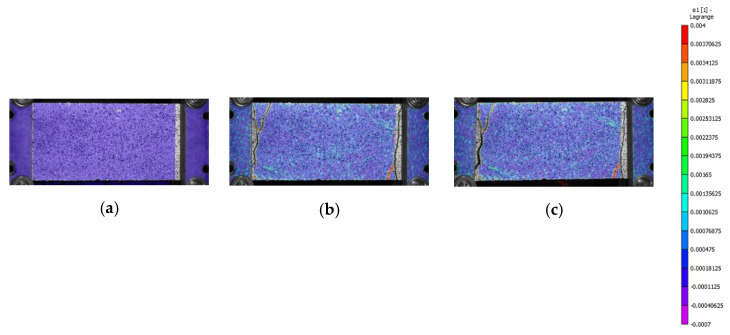
Pictures of specimen BP-2-A at the beginning of the test (**a**), maximum load (**b**) and after the drop in load (**c**). The color scale represents the grade of the major principal strains, from 0.004 units in red to −0.0007 units in purple.

**Figure 7 materials-16-01999-f007:**
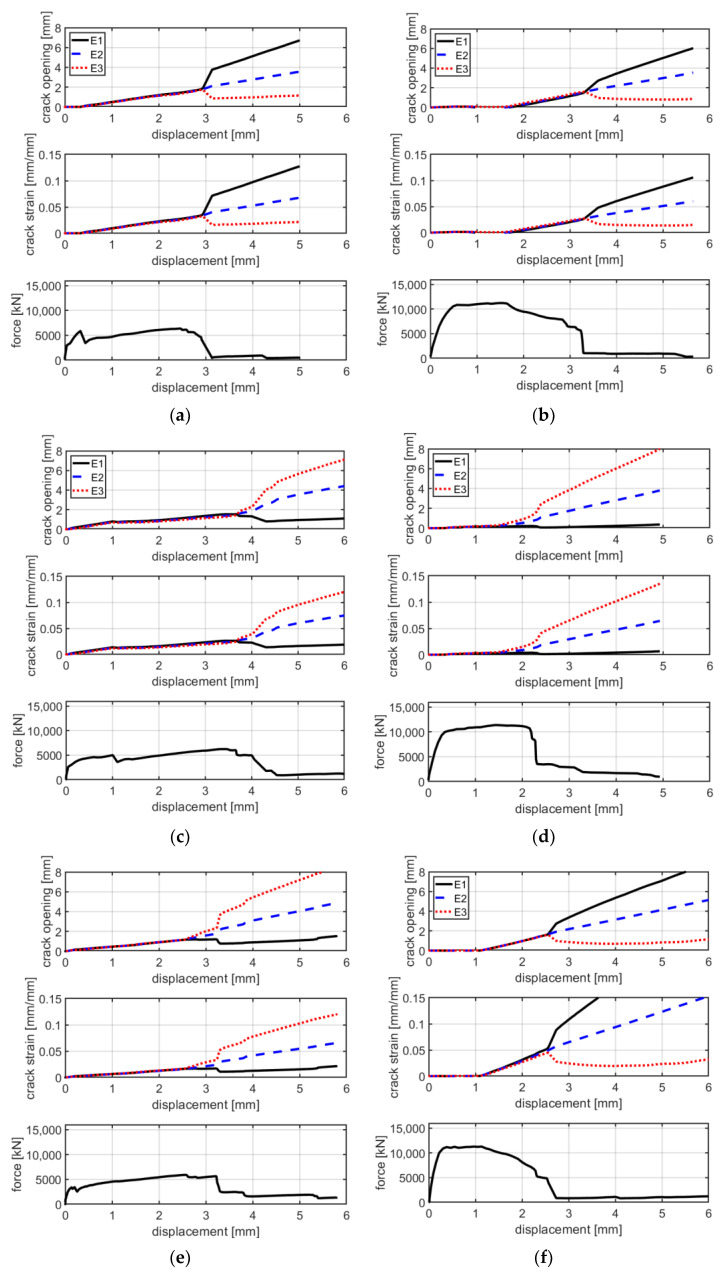
Comparison between crack opening, crack strain and axial force with the axial displacement of specimen BP-1-A (**a**), BP-2-A (**c**), BP-3-A (**e**), BPf-1-A (**b**), BPf-2-A (**d**) and BPf-3-A- (**f**).

**Figure 8 materials-16-01999-f008:**
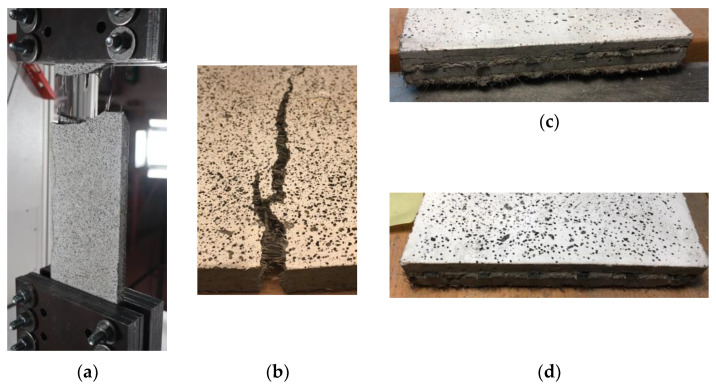
Failed specimens: (**a**) BP-1-A specimens, textile rupture; (**b**) main crack on specimen BPf-1-A and detail of the textile at the two ends of the specimen (**c**,**d**).

**Figure 9 materials-16-01999-f009:**
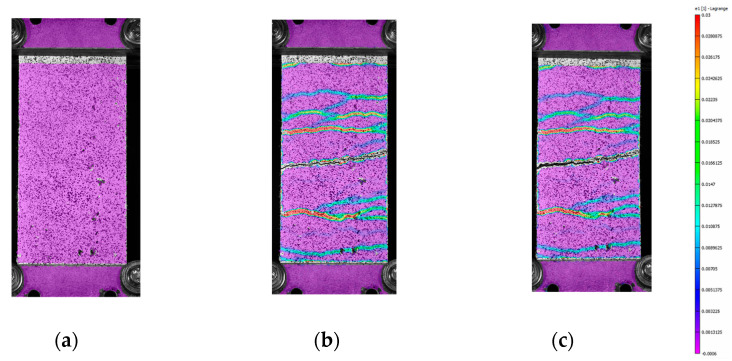
Crack pattern evolution of specimen BPf-3-A (**a**–**c**) during the tests. Pictures are taken at the beginning of the test (**a**), the end of the plateau (**b**) and immediately after the final drop in load (**c**). The scale of color represents the grade of the major principal strains, from 0.03 units in red to −0.0007 units in purple.

**Figure 10 materials-16-01999-f010:**
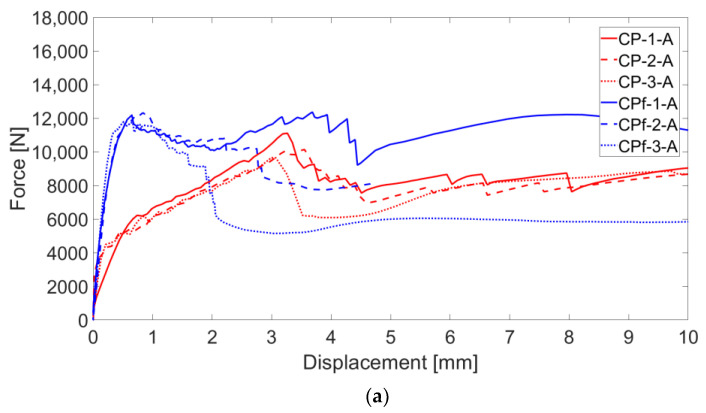
Force–displacement (**a**) and stress–strain (**b**) curves of C-TRC (in red) and C-F/TRC specimens (in blue).

**Figure 11 materials-16-01999-f011:**
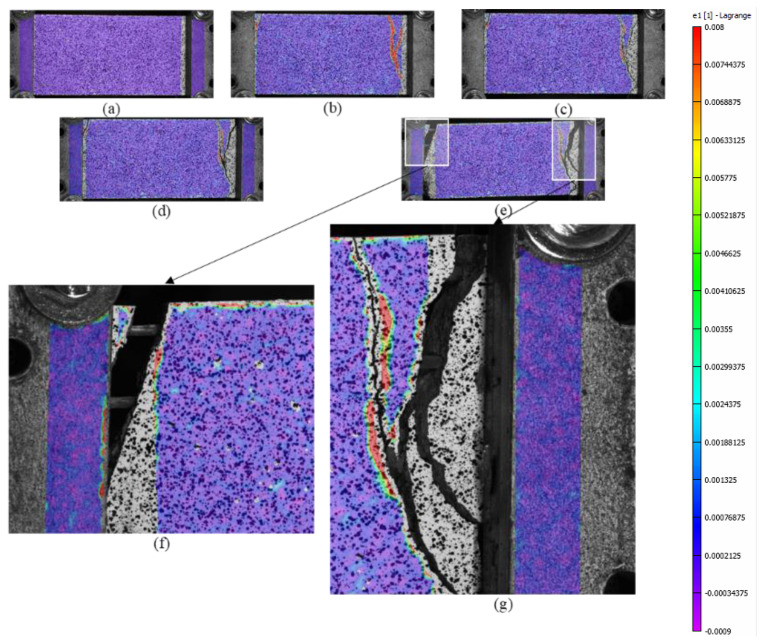
Pictures of specimen CP-1-A from DIC measurement during the various phases of the test: beginning of the test (**a**), appearance of the first crack (**b**), immediately before and after the first load drop (**c**,**d**), and when the test finished (**e**). Figures (**f**,**g**) are details of the cracks on the specimen. The color scale represents the grade of the major principal strains, from 0.008 units in red to −0.0009 units in purple.

**Figure 12 materials-16-01999-f012:**
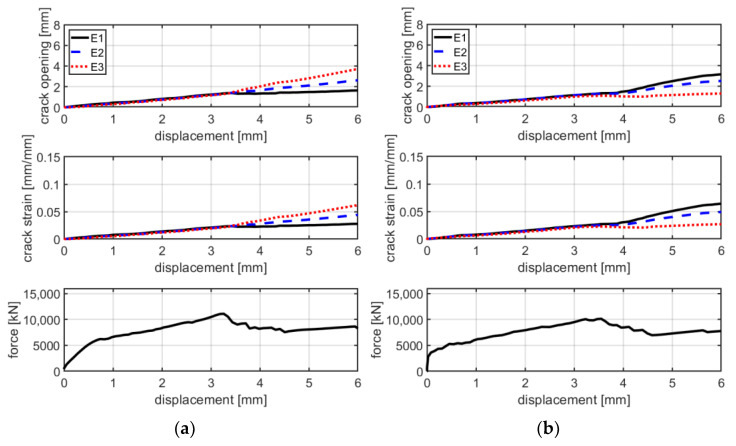
Crack opening/crack deformation/force–displacement curves for specimens CP-1-A (**a**), CP-2-A (**b**), CP-3-A (**c**), CPf-1-A (**d**), CPI-A (**e**) and CPf-3-A (**f**).

**Figure 13 materials-16-01999-f013:**
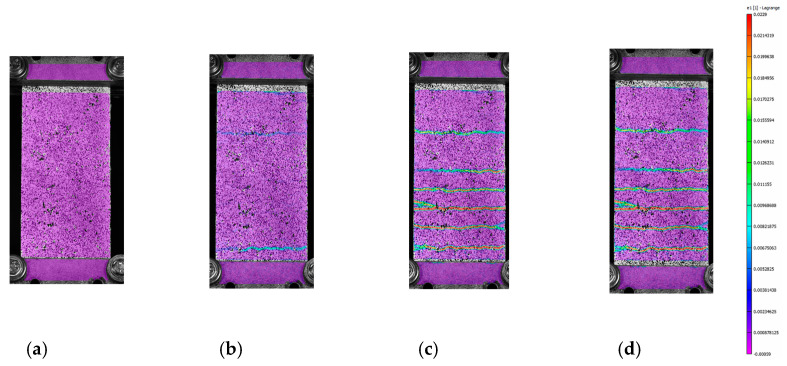
Frames of the DIC measurement of C-F/TRC specimen 3 at the beginning (**a**), the first crack (**b**), the end of stage 3 (**c**) and at the beginning of stage 4 (**d**). The color scale represents the grade of the major principal strains, from 0.0229 units in red to −0.00059 units in purple.

**Figure 14 materials-16-01999-f014:**
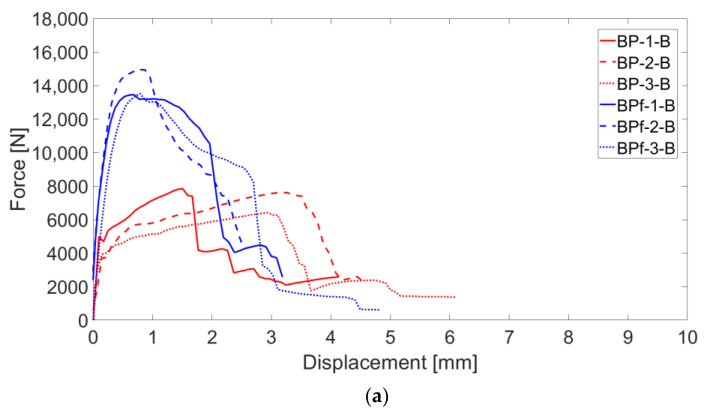
Force–displacement (**a**) and stress-strain (**b**) curves of B-TRC specimens (in red) and B-F/TRC specimens (in blue).

**Figure 15 materials-16-01999-f015:**
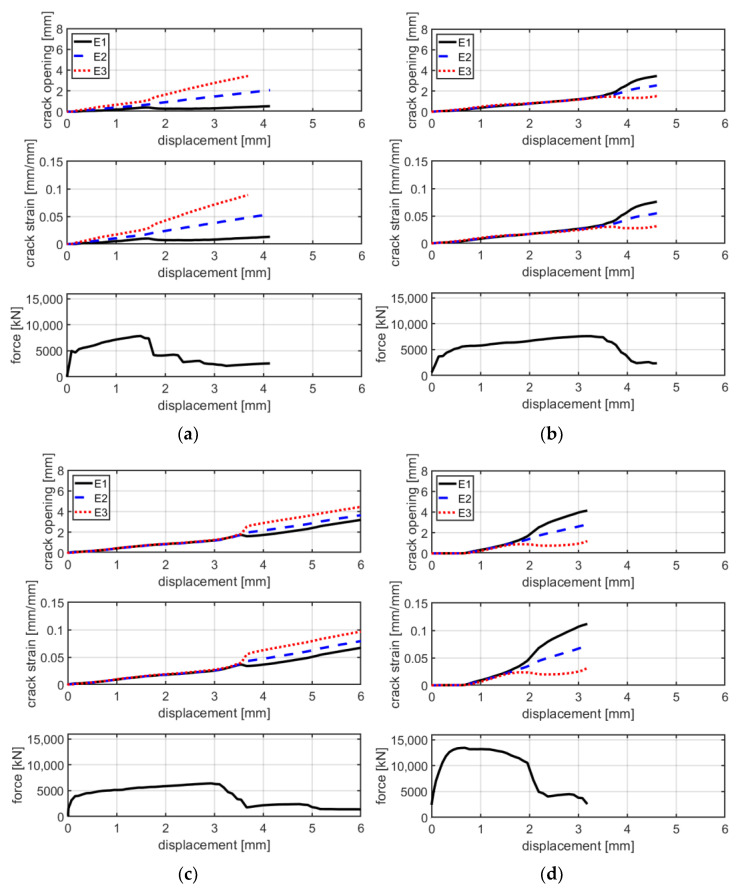
Graphs of crack opening/crack deformation/force-axial displacement of specimens BP-1-B (**a**), BP-2-B (**b**), BP-3-B (**c**), BPf-1-B (**d**), BPf-2-B (**e**) and BPf-3-b (**f**).

**Figure 16 materials-16-01999-f016:**
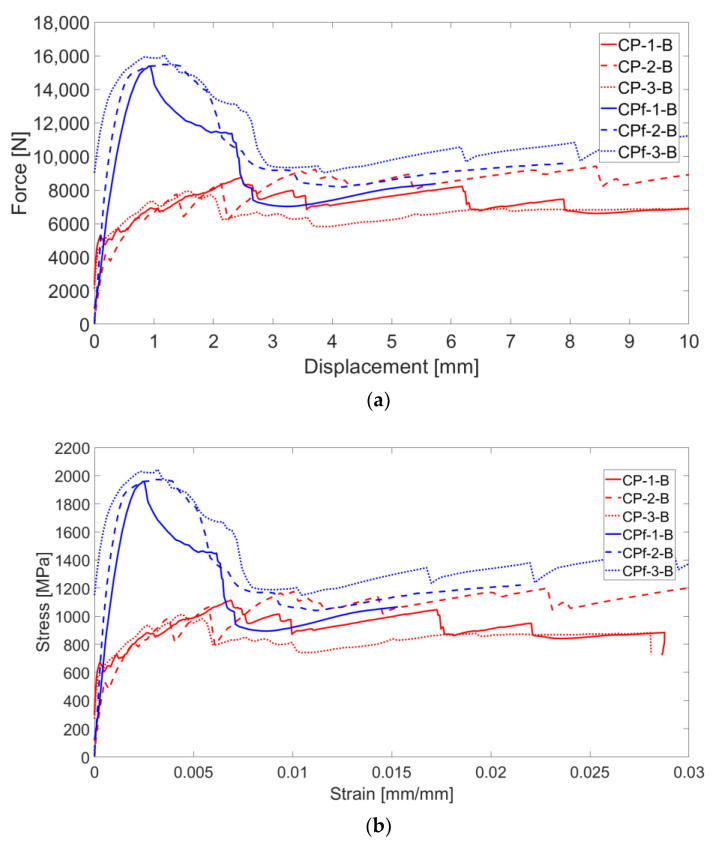
Force–displacement (**a**) and stress-strain (**b**) curves of C-TRC specimens (in red) and C-F/TRC specimens (in blue).

**Figure 17 materials-16-01999-f017:**
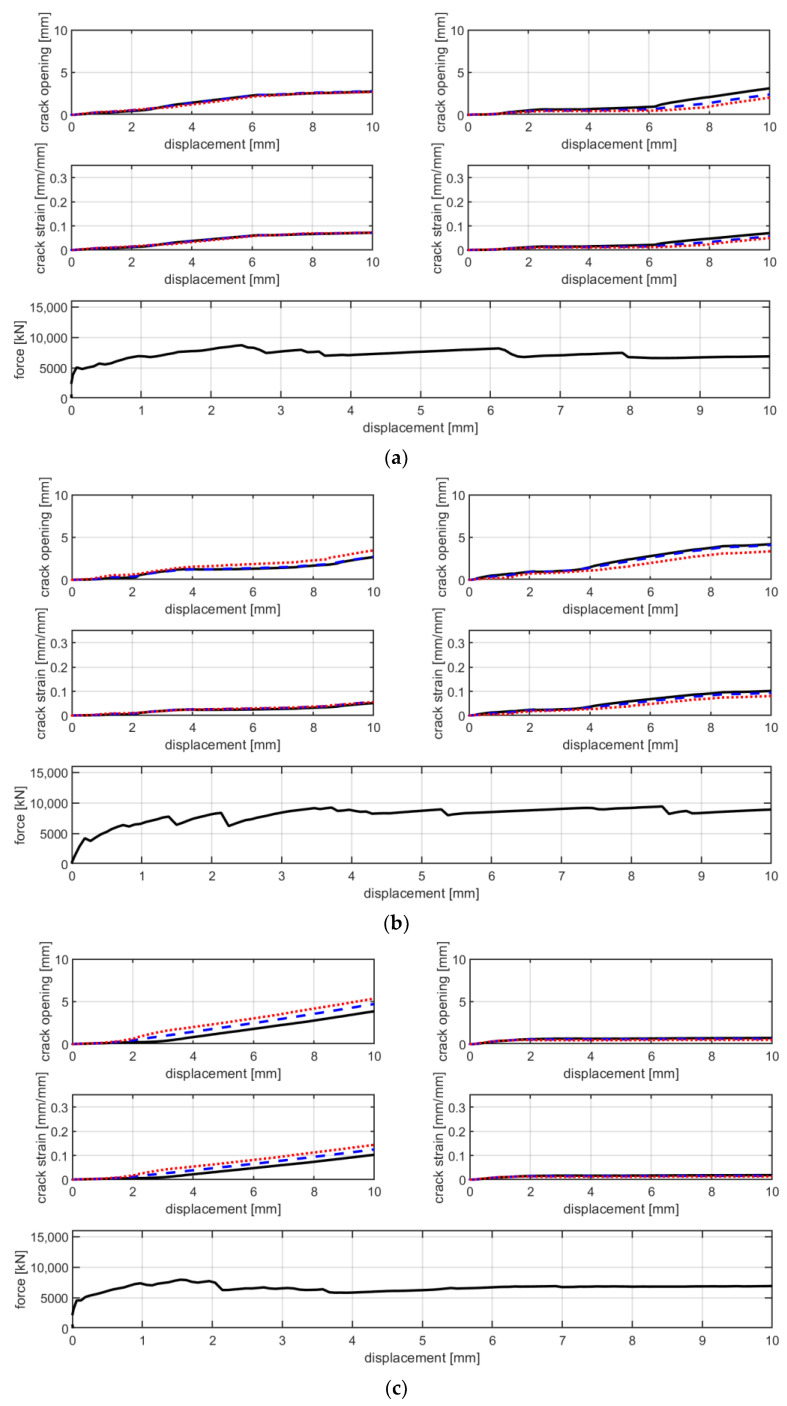
Graphs of crack opening/crack deformation/force-axial displacement of specimens CP-1-B (**a**), CP-2-B (**b**) and CP-3-B (**c**).

**Figure 18 materials-16-01999-f018:**
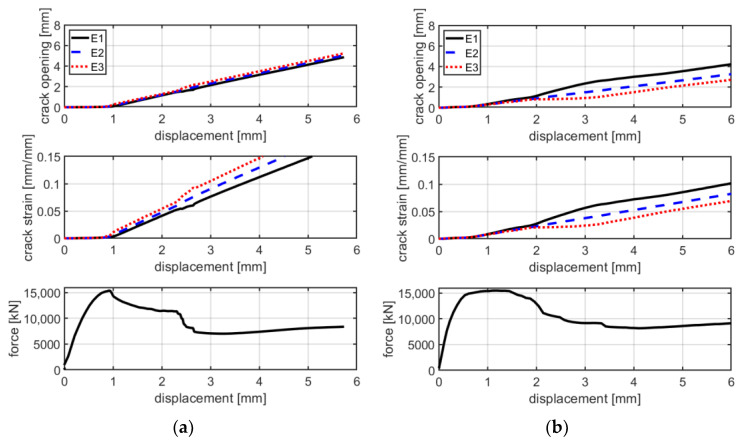
Graphs of crack opening/crack deformation/force-axial displacement of specimens CPf-1 (**a**), CPf-2 (**b**) and CPf-3 (**c**).

**Figure 19 materials-16-01999-f019:**
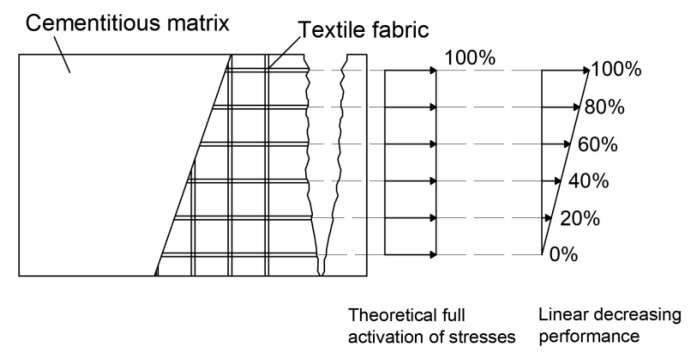
Theoretical and reduced distribution of stresses on basalt textile fabric.

**Figure 20 materials-16-01999-f020:**
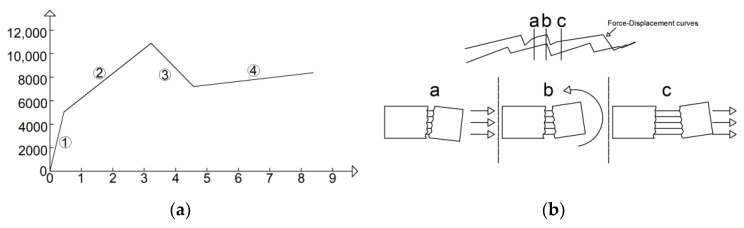
Simplified force–displacement curve of uniaxial tensile test on C-TRC specimens (**a**) and scheme of crack widening during the drop in load during stage 4 (**b**).

**Figure 21 materials-16-01999-f021:**
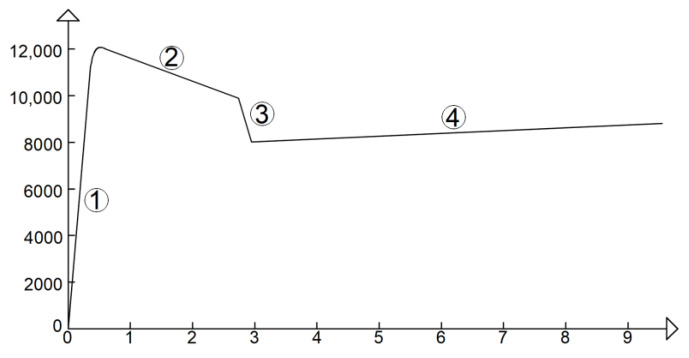
Simplified force–displacement curve of uniaxial tensile test on C-F/TRC specimens. The numbers 1–4 indicate the four stages of the curve.

**Table 1 materials-16-01999-t001:** Summary of the inorganic matrices and textile fabrics investigated in the literature.

Authors	Type of Inorganic Matrix	*f_c_*	Short Fibers	Textile Fabric
De Domenico et al. [6]	Cementitious matrix	22.03 29.30 19.87	Polymer	Basalt Carbon Steel
D’Antino and Papanicolaou [28]	Tixotropic fiber-reinforced cement-based matrix Lime-based mortar with silica sand Lime-based mortar with pozzolanic binders, synthetic fibers and graded sand	16.40 12.10 10.30		Carbon Basalt Glass Steel
De Felice et al. [29]	Pozzolanic-cement mortar Pozzolan lime-based mortar Cementitious mortar	37.0 13.0 22.6		Steel Carbon Basalt
Lignola et al. [30]	Cement based Lime based	25 12 15 15		Basalt
Larrinaga et al. [31]	Cement-based mortar	19.8		Basalt
Hojdys and Krajewski [27]	Cement based one-component mortar Cement-free NHL-based one-component mortar Natural hydraulic lime-based mortar Cement-based mortar	15.4 14.9 9.8 44.3	Synthetic fibers Fiber-reinforced	Carbon Glass PBO
Beßling et al. [32]	UHPC HPC	83 ± 7 60 ± 5		Carbon Basalt
Zhou et al. [33]	HPC	76.7	Steel	Carbon

**Table 2 materials-16-01999-t002:** Mechanical properties of the cementitious matrices.

	*f_c_* (MPa)	*f_t_* (MPa)
Cement matrix with short steel fibers	105.2	10.9
Cement matrix without short steel fibers	93.6	3.6

**Table 3 materials-16-01999-t003:** Mechanical and geometrical properties of textile fabrics.

	*g* (mm)	*A* (mm^2^/m)	*σ_t,tex_* Lengthways (MPa)	*σ_t,tex_* Crossways (MPa)	*ε_u_* Lengthways	*ε_u_* Crossways	*E* (GPa)
Carbon textile fabric	22	71	2531	2841	1.71%	1.47%	
Basalt textile fabric	20	65	1495	1495	1.61%	1.61%	92.7

**Table 4 materials-16-01999-t004:** Test results expressed in terms of average maximum load, equivalent yielding displacement, ultimate displacement and index of inelastic displacement.

Specimen ID	Maximum Load (N)	Average Maximum Load (N) (Stress (MPa))	CV Max Load	δ_y_ (mm)	δ_u_ (mm)	μ	μ Average	CV μ
BP-1-A	6364.5	6178.3 (792.1)	0.04	0.029	2.8	96.6	71.0	0.45
BP-2-A	6253.4			0.047	3.8	80.9		
BP-3-A	5916.9			0.090	3.2	35.6		
BPf-1-A	11,231	11,308 (1449.7)	0.01	0.65	2.2	3.38	4.4	0.21
BPf-2-A	11,386			0.45	2.2	4.9		
BPf-3-A	11,306			0.38	1.9	5		
CP-1-A	11,107	10,296.7 (1311.7)	0.07	0.06	3.76	62.7	71.5	0.40
CP-2-A	10,135			0.04	4.14	103.5		
CP-3-A	9648			0.07	3.38	48.3		
CPf-1-A	12,365	12,159 (1548.9)	0.03	0.57	4.43	7.77	5.65	0.38
CPf-2-A	12,315			0.49	2.78	5.67		
CPf-3-A	11,797			0.46	1.61	3.5		
BP-1-B	7848	7289.7 (934.6)	0.11	0.077	1.77	22.96	41.23	0.39
BP-2-B	7610			0.07	3.77	53.85		
BP-3-B	6411			0.07	3.28	46.89		
BPf-1-B	13,455	13,973.7 (1791.5)	0.06	0.78	1.96	2.51	8.46	1.28
BPf-2-B	14,944			0.63	1.22	1.94		
BPf-3-B	13,522			0.077	1.61	20.93		
CP-1-B	8708.9	8905.1 (1134.4)	0.12	0.18	6.37	35.39	18.18	0.84
CP-2-B	10,066			0.82	5.37	6.55		
CP-3-B	7940.6			0.17	2.14	12.59		
CPf-1-B	15,382	15,638.7 (1992.2)	0.02	0.88	1.54	1.75	3.2	0.47
CPf-2-B	15,482			0.69	2.14	3.11		
CPf-3-B	16,052			0.52	2.47	4.74		

## Data Availability

The data presented in this study are available from the corresponding author upon reasonable request.

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
