# Peer review of "Investigation of the Failure Modes of Textile-Reinforced Concrete and Fiber/Textile-Reinforced Concrete under Uniaxial Tensile Tests"

_materials, 2023, doi:10.3390/ma16051999_

Round 1

Reviewer 1 Report

In this paper, the failure modes of TRC and F/TRC under uniaxial tensile test were investigated. This research provided some insights into the combination of textile fabrics and cement-based matrix. However, great improvement is needed before it can be published. There are the following problems:

General comments:

(1) “Abstract” is very poor. The research background is too long, method and main findings are not adequate. Besides, the research significance and subsequent impact of this study on the state of the practice should be highlighted.

(2) “Keywords” are not accurate and comprehensive.

(3) Introduction: This part should be improved. Research background and research significance should be presented concisely and comprehensively in the first paragraph. Research on textile-reinforced concrete is quite common. The introductory part should be substantially improved providing more convincing motivations for this research. It is suggested to extend the literature review and make a table reporting literature with similar works.

(4) Conclusions: Many of the results and conclusions of this paper are quite basic (such as Conclusions 1, 5 and 6).

Specific comments:

(1) The reference format is not reasonable. Most of the statements in lines 26~44 should be referenced to enhance their reliability. Statements like “… control [1]–[15]” should be separated to avoid having too many citations at once.

(2) The information on similar research conditions should be compared with those in this study fully, thus the research gaps which the authors are addressing can be identified.

(3) The research method, content and significance should be put in a separate paragraph at the end of Introduction.

(4) Figure 2: Detailed descriptions on (a)~(e) are needed. Besides, Figures 6, 9, 11, 13~20 are of low quality and should be improved.

(5) Conclusions: 1) Many of the conclusions can be deleted or combined, no more than 5 of them can be reserved. 2) Qualitative statement like “short 475 steel fibres in the cementitious matrix of F/TRC specimens led to higher first crack stresses and higher tensile loads compared to those of TRC specimens” is not instructive, a similar conclusion is quite common. A quantitative description can be added. 3) The statement “Future studies are …” can not be a part of the conclusion. 4) The innovation should be highlighted in this part.

Author Response

The authors appreciate the comments and suggestions of the reviewers. The suggestions were found essential to the improvement of this manuscript. The comments have been answered below and implemented in the article (highlighted in yellow):

  1. “Abstract” is very poor. The research background is too long, method and main findings are not adequate. Besides, the research significance and subsequent impact of this study on the state of the practice should be highlighted.

The abstract has been improved due to the suggestions of the reviewer. (Please, see attached)

  1. “Keywords” are not accurate and comprehensive.

The keywords have been rewritten following the suggestions of the reviewer. (Please, see attached)

  1. Introduction: This part should be improved. Research background and research significance should be presented concisely and comprehensively in the first paragraph. Research on textile-reinforced concrete is quite common. The introductory part should be substantially improved providing more convincing motivations for this research. It is suggested to extend the literature review and make a table reporting literature with similar works.

The literature review has been improved and a table in which it was reassumed the literature review has been included as the reviewer suggested. The implementation concerns the article from line 72 to line 115. (Please, see attached)

Specific comments:

  1. The reference format is not reasonable. Most of the statements in lines 26~44 should be referenced to enhance their reliability. Statements like “… control [1]–[15]” should be separated to avoid having too many citations at once.

Reference format has been changed. Reference has been added to most of the statements, resulting in separating the too many citations at once. (Please, see attached ,lines 28-48)

  1. The information on similar research conditions should be compared with those in this study fully, thus the research gaps which the authors are addressing can be identified.

Study of similar researches and research gaps have been investigated from line 72 to line 115. (Please, see attached)

  1. The research method, content and significance should be put in a separate paragraph at the end of Introduction.

In order to follow the suggestion of the reviewer, the author created a new paragraph name “Research method, content and significance”. (Please, see attached, lines 116-133).

  1. Figure 2: Detailed descriptions on (a)~(e) are needed. Besides, Figures 6, 9, 11, 13~20 are of low quality and should be improved.

The text between line 205 and 220 have been improved. Figures 6, 9, 11, 13~20 have been realized with bigger text and curves. (Please, see attached)

  1. Conclusions: 1) Many of the conclusions can be deleted or combined, no more than 5 of them can be reserved. 2) Qualitative statement like “short 475 steel fibres in the cementitious matrix of F/TRC specimens led to higher first crack stresses and higher tensile loads compared to those of TRC specimens” is not instructive, a similar conclusion is quite common. A quantitative description can be added. 3) The statement “Future studies are …” can not be a part of the conclusion. 4) The innovation should be highlighted in this part.

Conclusions have been improved on the basis of the suggestions of the reviewer. (Please, see attached lines 639-670)

Reviewer 2 Report

Manuscript Number: materials-2149835

Full Title: Investigation on failure modes of TRC and F/TRC under uniaxial tensile test

In this manuscript, the results of an experimental investigation of TRC and F/TRC composites are presented. The manuscript's data are of interest to the journal's broad readership, especially those seeking innovative materials for structural rehabilitation and retrofitting purposes. The paper is well-written and well-structured and presents important data so I would recommend its publication after some minor comments given hereafter.

Comments:  

·         In Table 1 and figures 3, 9, and 15, I suggest representing data in terms of averaged stress and strain rather than load and displacement. This will make more sense for readers keen to compare this innovative material's (averaged) properties with conventionally used retrofitting materials.

·         It is recommended to add a legend (if possible) to the images acquired from the virtual extensometer.

·         The volume(weight) fraction of steel fibers should be mentioned.

Author Response

The authors appreciate the comments and suggestions of the reviewers. The suggestions were found essential to the improvement of this manuscript. The comments have been answered below and implemented in the article (highlighted in yellow):

  1. In Table 1 and figures 3, 9, and 15, I suggest representing data in terms of averaged stress and strain rather than load and displacement. This will make more sense for readers keen to compare this innovative material's (averaged) properties with conventionally used retrofitting materials.

Values of average maximum stress has been included in Table 1, furthermore, it has been added stress-strain curves of the specimens below the force-displacement curves (Please see attached, pages 8 9,14, 18, and 21) .

  1. It is recommended to add a legend (if possible) to the images acquired from the virtual extensometer.

Legend has been inserted next to the pictures (please see attached, pages 10, 13, 15, 17).

  1. The volume(weight) fraction of steel fibers should be mentioned.

The information concerning the volume fraction of short steel fibres admixed in the cementitious matrix has been included in the paragraph “Materials and Method’ in the line 138:“An amount of 2.5 Vol.-% of short steel fibres were admixed in the cementitious matrix.”.(Please, see attached)

Reviewer 3 Report

The article deals with the experimental analysis of the investigation of failure modes of TRC and F/TRC under tensile loading conditions. The manuscript is well written, and the reviewer appreciates the authors' effort in conducting the experimental analysis. However, the novelty is not strong, and the mechanism is not revealed in-depth enough, which needs to be combined with more in-depth microscopic testing. So, it may not be suitable for publication.

The reviewer has a few comments and suggestions on the article as below:

1. The authors may avoid using abbreviations in the article's title.

2. The novelty of the paper should be highlighted in detail in the introduction, and the Motivation of the research significance is not reported. The authors have to explain what is new here compared to the previous studies.

3. Try to mention a problem that needs solving - in other words, clarify the research question underlying your study.

4. The abstract needs to focus on the effective contribution of this work and more explanation of the outcomes temporarily. Since there are already many works in this area.

5. The materials and methods section need to be improved since there is no clarity in the section. The properties of the materials can be tabulated for better understanding.

6. The experimental setup and details about the measuring instruments need to be included in a separate section. Like, how the crack opening, crack deformation, and displacement were measured during the experiment)

7. The discussion section must be included a detailed analysis of the behaviour of the specimens and what is new here compared with the previous studies.

8. Figure 1 needs to be included with labels, and the units of the dimension need to be included.

9. Check the English language and grammar in the manuscript. 

Author Response

The authors appreciate the comments and suggestions of the reviewers. The suggestions were found essential to the improvement of this manuscript. The comments have been answered below and implemented in the article (highlighted in yellow):

  1. The authors may avoid using abbreviations in the article's title.

The title is changed as suggested by the reviewer: “Investigation of the failure modes of Textile Reinforced Concrete and Fibre/Textile Reinforced Concrete under uniaxial tensile test”. (Please, see attached)

  1. The novelty of the paper should be highlighted in detail in the introduction, and the Motivation of the research significance is not reported. The authors have to explain what is new here compared to the previous studies.

The introduction, between line 72 and line 125, has been improved by following the suggestions of the reviewer. (Please, see attached)

  1. Try to mention a problem that needs solving - in other words, clarify the research question underlying your study.

The authors improved the reason and necessity of this research in the paragraph “Research method, content and significance” of the article. (Please, see attached, lines 117-125)

  1. The abstract needs to focus on the effective contribution of this work and more explanation of the outcomes temporarily. Since there are already many works in this area.

The abstract has been improved by following the suggestion of the reviewer. (Please, see attached, lines 11-23)

  1. The materials and methods section need to be improved since there is no clarity in the section. The properties of the materials can be tabulated for better understanding.

The paragraph “Materials and methods” has been improved, properties of materials are now listed in table 2 and table 3 as suggested by the reviewer (Please, see attached, lines 152-155).

  1. The experimental setup and details about the measuring instruments need to be included in a separate section. Like, how the crack opening, crack deformation, and displacement were measured during the experiment)

The details about the test setup are now included in the article between line 190 and 194 and between 207 and 218. Furthermore, in figure 3 details of the test setup are shown. (Please, see attached)

  1. The discussion section must be included a detailed analysis of the behaviour of the specimens and what is new here compared with the previous studies

From line 600 to line 620 the discussion section has been improved by following the suggestions of the reviewer. (Please, see attached)

  1. Figure 1 needs to be included with labels, and the units of the dimension need to be included.

Figure 1 has been improved as suggested by the reviewer. (Please, see attached)

  1. Check the English language and grammar in the manuscript.

The manuscript has been submitted to the supervision of the Language Editing Services. Language and grammar of the manuscript are improved. (Please, see attached)

Reviewer 4 Report

The manuscript presents the tensile behavior Fiber/Textile-Reinforced Concrete (F/TRC) composites. A series of direct tensile tests were performed to characterize the tensile properties of F/TRC with different textile types (carbon and basalt) and short steel fiber. Generally, the paper was organized well. There are, however, some points in the text that could be improved. The Authors are encouraged to respond to the questions raised below before making a resubmission:

Comment 1: Section 2: The tensile properties of steel fiber (ultimate tensile stress and strain, as well as elastic modulus) should be reported.

Comment 2: Lines 89-94: Please add also the elastic modulus and the ultimate strain of textiles (carbon and basalt)

Comment 3: Lines 97-98: The text (“These specimens … of the layer”) should be revised.

Comment 4: Lines 104-107: The text (“The overlap is … respectively”) should be revised.

Comment 5: Figure 1: In series B details, it is suggested to use a darker color only in the areas where textiles overlap.

Comment 6: Section 2: A figure showing the test setup should be added.

Comment 7: Lines 136-137: Why was virtual extensometer used?

Comment 8: In table 1: The coefficient of variation should also be added for the average values. In addition, the authors should provide a comment on why some specimens (BP-A, CP-A, CPf-A, BP-B, BPf-B, CP-B, and CPf-B) have high Coefficients of Variation of inelastic displacement.

Comment 9: Figure 4: Does this figure shows the failure mode of BP-A specimen of BPf-A?

Comment 10: Figure 6: Please change the crack deformation to crack strain with its unit (mm/mm or %).

Comment 11: Figure 6: This figure presents 18 subfigures, while there is only 4 supportive sentences. The authors should explain more about the curves.

Comment 12: Figures should be presented in the order they appear in the text. For example, on page 8, Figure 8 is first introduced in the text (line 187), then Figure 7 appears (line 195).

Comment 13: Lines 206-207: Since the effective bond length of basalt textile is almost 25~30 mm and the gripping length was equal to 180 mm (as mentioned in line 126), how was the textile pulled out from the matrix? The authors should add comments about this observation.

Comment 14: Figures 11 and 14, 16, and 17: Please see comment 11.

Comment 15: Authors need to clearly state the limitations of this study.

Author Response

The authors appreciate the comments and suggestions of the reviewers. The suggestions were found essential to the improvement of this manuscript. The comments have been answered below and implemented in the article (highlighted in yellow):

Comment 1: Section 2: The tensile properties of steel fiber (ultimate tensile stress and strain, as well as elastic modulus) should be reported.

The fibers are type Weidacon FM provided by Stratec company. Specific mechanical properties were not provided by the company, but the failure mode was due to the short length always pull-out so that such properties are in this special case not very relevant.

Comment 2: Lines 89-94: Please add also the elastic modulus and the ultimate strain of textiles (carbon and basalt)

The ultimate strain, provided by the producer, has been included in the article. The elastic modulus of the basalt textile fabric has also been reported, however, the elastic modulus of the carbon textile fabric is not reported in its technical sheet. (Please, see attached)

Comment 3: Lines 97-98: The text (“These specimens … of the layer”) should be revised.

The text has been revisited has suggested by the reviewer. (Please, see attached, lines 158-160)

Comment 4: Lines 104-107: The text (“The overlap is … respectively”) should be revised.

The text has been revisited has suggested by the reviewer. (Please, see attached, lines 166-168).

Comment 5: Figure 1: In series B details, it is suggested to use a darker color only in the areas where textiles overlap.

The figure has been improved with the suggestions of the reviewer. (Please, see attached, page 5)

Comment 6: Section 2: A figure showing the test setup should be added.

The authors included two pictures of the test setup. (Please, see attached, page 7)

Comment 7: Lines 136-137: Why was virtual extensometer used?

It was used to measure the imposed displacement of the two clamps of test machine during the test. This measurement is used to plot force-displacement curves and crack opening-displacement curves. This information has been included in the article between line 207 and line 217 (please, see attached):

“Through the analysis with the DIC software it is possible to select two points in the selected area and to measure the displacement of these points during the test. These points together are referred to in this article as the “virtual extensometer”. The axial displacement imposed by the test machine during the test is measured through a virtual extensometer by placing two points in the middle of the two steel clamps, one on the top and one on the bottom of the test machine. The measurement of crack opening and deformation are obtained by placing three virtual extensometers across the crack that propagate in te specimen. Therefore, the two virtual points are placed on the right and on the left of the measured crack. Crack deformation values are obtained by dividing the crack displacement by the initial displacement.”

Comment 8: In table 1: The coefficient of variation should also be added for the average values. In addition, the authors should provide a comment on why some specimens (BP-A, CP-A, CPf-A, BP-B, BPf-B, CP-B, and CPf-B) have high Coefficients of Variation of inelastic displacement.

The author included in the table two more columns with Coefficient of Variation of maximum load and index of inelastic displacement. It has also been included an explanation about the high values of Coefficient of Variation of μ. (Please see attached, lines 227-239 and table 4 page 9)

Comment 9: Figure 4: Does this figure shows the failure mode of BP-A specimen of BPf-A?

The figure refers to the failure of a BP-A specimen. This specification has been included in the image caption. (Please, see attached, page 10)

Comment 10: Figure 6: Please change the crack deformation to crack strain with its unit (mm/mm or %).

Crack deformation has been changed in crack strain [mm/mm]. (Please see attached, pages 10, 11, 15,16, 19, 20, 22, ans 23)

Comment 11: Figure 6: This figure presents 18 subfigures, while there is only 4 supportive sentences. The authors should explain more about the curves.

Discussion about the curves has been expanded in figure 6, figure 11, figure 14, figure 16, and figure 17 (Please, see attached lines 267-279, 281-283, 314-317, 345-347, 368-370, 392-396, 427-433, 448-450).

Comment 12: Figures should be presented in the order they appear in the text. For example, on page 8, Figure 8 is first introduced in the text (line 187), then Figure 7 appears (line 195).

The position of figures has been changed. Figures are in the same order has they are presented.

Comment 13: Lines 206-207: Since the effective bond length of basalt textile is almost 25~30 mm and the gripping length was equal to 180 mm (as mentioned in line 126), how was the textile pulled out from the matrix? The authors should add comments about this observation.

In the manuscript it is mentioned the overlap length of the textile fabric. This measure, of 250 mm for the basalt textile, is the length in which two different pieces of textile fabric are overlapped in order to avoid premature failure of the specimens. In the case of specimens with basalt textile, as showed in figure 8, the failure is due to the rupture of the textile fabric and not to the pull out of the textile fabric.

Comment 14: Figures 11 and 14, 16, and 17: Please see comment 11.

Discussion about the curves has been expanded in figure 6, figure 11, figure 14, figure 16, and figure 17. (Please, see attached)

Comment 15: Authors need to clearly state the limitations of this study.

Limitations of this study have been included between lines 625 and 628. (Please, see attached)

Reviewer 5 Report

The research deals with an experimental investigation about the performances of TRC and F/TRC composites through uniaxial tensile test

Stress-strain curves of the coupons are missing along the manuscript. This type of curves could be more useful than the force-displacement ones.

A big description and An image of the test setup should appear in the paper.

The colours of Figures 4, 5, 7, 10, 12 represent a magnitude, nevertheless neither the magnitude nor the value scale appear in those figures.

Line 321: “one side of the crack tends to widen more, resulting in a “rotation” of the specimen during the test”. Could this fact happen because of an incorrect test setup? Did the test setup was performed according to a code or official recommendation? This information should appear in the paper.

Author Response

The authors appreciate the comments and suggestions of the reviewers. The suggestions were found essential to the improvement of this manuscript. The comments have been answered below and implemented in the article (highlighted in yellow):

  1. Stress-strain curves of the coupons are missing along the manuscript. This type of curves could be more useful than the force-displacement ones.

Stress-strain curves has been included in the manuscript. (Please, see attached pages 9, 14, 18, and 21)

  1. A big description and An image of the test setup should appear in the paper.

An image and a description of the test setup has been included. Between lines 190 and 198 and figure 3 the test setup is described. (Please see attached)

  1. The colours of Figures 4, 5, 7, 10, 12 represent a magnitude, nevertheless neither the magnitude nor the value scale appear in those figures.

Due to the suggestions of the reviewer the scale of colours have been included in the article, next to the figures mentioned by the reviewer. (Please, see attached at pages 10, 13, 15,and 17)

  1. Line 321: “one side of the crack tends to widen more, resulting in a “rotation” of the specimen during the test”. Could this fact happen because of an incorrect test setup? Did the test setup was performed according to a code or official recommendation? This information should appear in the paper.

The dimensions of the specimens and how the specimens is clamped to the test machined followed the recommendations of RILEM TC 232-TDT. The load was transferred to the specimen by friction. A thin sheet of rubber was positioned between the steel plates and the specimen. Therefore, the pressure of the steel plates to the specimens was adjusted to avoid slippage between the clamp and the specimens. Eventually slippage between clamp and specimen has not been observed, hence the author would not consider such failure related to an incorrect setup. Information about the test setup has been improved in the article in the lines 190-194. (Please, see attached)

Round 2

Reviewer 1 Report

The corresponding author in the website is not the one listed in the manuscript. Besides, the revision mode PDF version of the revised manuscript is very chaotic and does not respond point by point to the comments. 

Author Response

The authors appreciate the comments and suggestions of the reviewers. The suggestions were found essential for the improvement of this manuscript. Please see the attachment in which a response point by point is presented.

Reviewer 3 Report

The author carefully revised all the questions.

Author Response

The authors appreciate the reviewers' comments and suggestions, which are essential to the quality of the manuscript. The authors have read the reviewer's final comment.

Reviewer 4 Report

  • Section 3: The tensile properties of steel fiber (ultimate tensile stress and strain, as well as elastic modulus) should be reported.
  • The authors should comment on why some specimens (BP-A, CP-A, CPf-A, BP-B, BPf-B, CP-B, and CPf-B) have high Coefficients of Variation of inelastic displacement (Table 4).
  • Figure 7 (also figures 12, 15, 17, and 18) presents 18 subfigures, with only 4 supportive sentences. The authors should explain more about the curves.
  • Lines 326-329: Since the effective bond length of basalt textile is almost 25~30 mm, and the gripping length was equal to 180 mm (as mentioned in line 126), how was the textile pulled out from the matrix? The authors should add comments about this observation.
  • The authors should present an answer sheet and highlight the changes in the manuscript.

Author Response

(The authors gave the same response as above.)

Reviewer 5 Report

The authors addressed all reviewer comments. Therefore, the paper is suitable for publication.

Author Response

(The authors gave the same response as above.)
